# Silencing the *CsSnRK2.11* Gene Decreases Drought Tolerance of *Cucumis sativus* L.

**DOI:** 10.3390/ijms242115761

**Published:** 2023-10-30

**Authors:** Peng Wang, Zilong Wan, Shilei Luo, Haotai Wei, Jianuo Zhao, Guoshuai Wang, Jihua Yu, Guobin Zhang

**Affiliations:** 1State Key Laboratory of Aridland Crop Science, Gansu Agricultural University, Lanzhou 730070, China; wangpeng@st.gsau.edu.cn (P.W.); wanzl@st.gsau.edu.cn (Z.W.); luosl1021@163.com (S.L.); weiht@st.gsau.edu.cn (H.W.); zhaojn@st.gsau.edu.cn (J.Z.); wanggs@st.gsau.edu.cn (G.W.); yujihuagg@163.com (J.Y.); 2College of Horticulture, Gansu Agricultural University, Lanzhou 730070, China

**Keywords:** VIGS, drought stress, photosynthesis, antioxidant, transcriptome

## Abstract

Drought stress restricts vegetable growth, and abscisic acid plays an important role in its regulation. Sucrose non-fermenting1-related protein kinase 2 (SnRK2) is a key enzyme in regulating ABA signal transduction in plants, and it plays a significant role in response to multiple abiotic stresses. Our previous experiments demonstrated that the *SnRK2.11* gene exhibits a significant response to drought stress in cucumbers. To further investigate the function of *SnRK2.11* under drought stress, we used VIGS (virus-induced gene silencing) technology to silence this gene and conducted RNA-seq analysis. The *SnRK2.11*-silencing plants displayed increased sensitivity to drought stress, which led to stunted growth and increased wilting speed. Moreover, various physiological parameters related to photosynthesis, chlorophyll fluorescence, leaf water content, chlorophyll content, and antioxidant enzyme activity were significantly reduced. The intercellular CO_2_ concentration, non-photochemical burst coefficient, and malondialdehyde and proline content were significantly increased. RNA-seq analysis identified 534 differentially expressed genes (DEGs): 311 were upregulated and 223 were downregulated. GO functional annotation analysis indicated that these DEGs were significantly enriched for molecular functions related to host cells, enzyme activity, and stress responses. KEGG pathway enrichment analysis further revealed that these DEGs were significantly enriched in phytohormone signalling, MAPK signalling, and carotenoid biosynthesis pathways, all of which were associated with abscisic acid. This study used VIGS technology and transcriptome data to investigate the role of *CsSnRK2.11* under drought stress, offering valuable insights into the mechanism of the SnRK2 gene in enhancing drought resistance in cucumbers.

## 1. Introduction

Cucumber (*Cucumis sativus* L.), one of the most commonly cultivated vegetables worldwide, constitutes a large proportion of the vegetable crops in China. Cucumbers inevitably encounter adverse biotic and abiotic environmental conditions during their growth and development. Among these abiotic stressors, drought causes the greatest economic loss to plants [1]. When plants are subjected to drought, plant leaf morphological structure, physiological metabolism, drought-resistant gene function, and proteome expression are altered to varying degrees [2,3]; the content of abscisic acid (ABA) in plants also increases, which activates the ABA hormone signalling pathway; stress signals are recognised and transmitted to different cells through specialised signalling pathways, of which protein phosphorylation catalysed by protein kinases is a key component [4,5].

Plant SnRKs are classified into the SNF1/AMPK family. AMPK was originally defined as a mammalian protein kinase capable of phosphorylating and inactivating lipid and cholesterol synthesis. SNF1 kinases have characteristics similar to those of AMPK, including kinase structure, substrate specificity, mode of regulation, cellular metabolism, and response to environmental stress [6]. SnRKs are AMPK/SNF1 homologues that include two typical structural domains: the N-terminal catalytic domain and the C-terminal regulatory region. Based on the conserved nature of the kinase active domain, it can be divided into three subfamilies: SnRK1, SnRK2, and SnRK3. SnRK2 was initially thought to be a kinase involved in plant responses to abiotic stressors; subsequently, members of the SnRK2 subfamily were considered major players in plant responses to osmotic stress and ABA-dependent environmental signals [7]. The ABA signalling pathway is a key pathway in plants’ response to adversity. SnRK2 is involved in ABA signalling, forming the ABA–PYR–PP2C–SnRK2 downstream transcription factor-coupled signalling pathway.

Studies have shown that transgenic *Arabidopsis* overexpressing *TaSnRK2.3* showed no significant differences in water retention ability (WRA), free proline content, and chlorophyll from the wild-type under normal conditions. Under drought stress, transgenic plants showed higher WRA, proline, and chlorophyll contents than wild-type plants, the main roots of the former were longer, and lateral roots were more abundant than those of the wild-type [8]. This finding indicates that *TaSnRK2.3* is a multifunctional regulator with potential applications in transgenic breeding to improve drought tolerance in crops. The literature found that transgenic plants overexpressing *OsSAPK9* had improved physiological indicators such as water retention capacity, soluble sugar, proline content, stomatal closure, membrane stability, cell detoxification, and enhanced drought resistance [9]. A similar phenotype was observed in transgenic poplars overexpressing *AtSnRK2.8* [10]. In transgenic *tobacco* overexpressing *SoSnRK2.1*, under drought stress, ionic liquid (IL), MDA, hydrogen peroxide (H_2_O_2_), superoxide dismutase (SOD), peroxidase (POD), and catalase (CAT) activities increased [11], indicating that overexpression of *SoSnRK2.1* enhanced the drought tolerance of *tobacco*. Another study found that the *Fagopyrum tataricum* transcription factor FtbZIP5 was regulated by *FtSnRK2.6* to enhance drought tolerance in transgenic *Arabidopsis* [12]. Katsuta et al. [13] observed that the abiotic stress response of RAF kinase plays a critical role in the SnRK2-mediated osmotic stress response by interacting with *AtSnRK2.6*, a central factor in stomatal closure under drought conditions, to protect plants from drought injury. These findings suggest that SnRK2 plays a crucial role in regulating plant responses to drought stress.

Virus-induced gene silencing (VIGS) is a genetic technique that inhibits the expression of endogenous genes in plants by inserting recombinant viruses into target gene fragments and is used mainly for the functional analysis of genes [14]. The VIGS system has been reported in cucumbers. PDS and SU were cloned from cucumber leaves, and two vectors, ALSV-PDS and ALSV-SU, were constructed to infect cucumbers. The plants exhibited a bleaching phenotype, and a cucumber-ALSV-mediated VIGS system was successfully constructed [15]. Another study utilised TRV (tobacco rattle virus; it is a widely used, efficient, and persistent viral vector that can mediate gene silencing without causing virus-induced symptoms) virus mediated construction of a TRV2-*CuGPAT6* vector, silencing cucumber plants. The silencing strain enhanced its autotoxicity to cinnamic acid, and successfully constructed a cucumber-TRV-mediated VIGS system [16]. Previous studies have shown that VIGS technology provides a new approach and an effective and efficient means for studying cucumber gene function.

However, little research has been conducted on the *SnRK2* gene in cucumber, especially on the role of *SnRK2* in drought resistance. Our previous experiments showed that *CsSnRK2.11* was significantly upregulated by drought stress [17]. In this experiment, we utilise VIGS technology to study the effects of silencing *CsSnRK2.11* gene on cucumber plants under drought stress at physiological and biochemical levels; we combine transcriptome data to analyse its function under drought stress, providing insights into the mechanism by which *SnRK2* gene enhances cucumber drought resistance.

## 2. Results

### 2.1. Plant Phenotype and Relative Expression Level of CsSnRK2.11 under Drought Stress

On the first day of the drought treatment, there was no significant difference in wilting among the treatments (Figure 1). The plant phenotype and root size of pTRV2-0 and pTRV2-*SnRK2.11* plants were smaller than those of CK plants on the 7th day after stopping watering, all seedlings showed wilting, and their growth was severely inhibited. The pTRV2-*SnRK2.11* plants had the most obvious degree of wilting and the lowest growth rate, followed by the pTRV2-0 plants. Regarding root growth, there were significant differences among drought stress, pTRV2-0, pTRV2-*SnRK2.11*, and CK plants’ aboveground parts, and all roots showed yellowing. The lateral root content of drought stress and pTRV2-0 plants was greater than that of CK plants, but the primary root length was less than that of CK plants. The root system of pTRV2-*SnRK2.11* plants was most severely affected by drought stress, and its root length and lateral root number were the least. The *SnRK2.11* gene expression level of drought stress plants significantly increased, reaching more than five times that of CK plants. The *SnRK2.11* gene expression level of pTRV2-*SnRK2.11* plants was less than 50% of that of CK plants, indicating the target gene *SnRK2.11* was almost not expressed under drought stress after silencing.

### 2.2. Effect of Drought Stress on Chlorophyll Content, Relative Water Content, and Relative Electrical Conductivity of Plants

As shown in Figure 2, The Chl a, Chl b, total chlorophyll, and carotenoid contents of drought stress, pTRV2-0, and pTRV2-*SnRK2.11* plants were significantly lower than and significantly different from those of CK plants (Figure 2A–D). The pTRV2-*SnRK2.11* plants had the lowest Chl a, Chl b, total chlorophyll, and carotenoid contents, 38.5%, 30.5%, 36.6%, and 36.4% lower than those of CK plants, respectively, indicating that silencing the *SnRK2.11* gene had the least effect on Chl b content and the greatest effect on Chl a, total chlorophyll, and carotenoid contents. The relative leaf water content of drought stress, pTRV2-0, and pTRV2-*SnRK2.11* plants was significantly lower than that of CK plants, and was 9%, 8.4%, and 14.33% lower than that of CK plants, respectively. (Figure 2E). The difference in relative leaf water content between drought stress and pTRV2-0 plants was not significant. In addition, the relative conductivities of drought stress, pTRV2-0, and pTRV2-*SnRK2.11* plants were significantly greater than those of CK plants and were 19.4%, 7%, and 9.3% lower than that of CK plants, respectively (Figure 2F). These results indicated that silencing *CsSnRK2.11* led to an increase in membrane permeability and an increase in the degree of membrane damage.

### 2.3. Effect of Drought Stress on Gas Exchange Parameters and Chlorophyll Fluorescence Parameters of Plant Leaves

Photosynthetic gas exchange parameters were used to measure the photosynthetic activity of the plant leaves. The Pn, Tr, and Gs values in drought stress, pTRV2-0, and pTRV2-*SnRK2.11* plants were significantly lower than those of CK plants; pTRV2-*SnRK2.11* plants had the lowest values: 61.7%, 70.3%, and 75.5% lower than those of CK plants (Figure 3A–C). The Ci in the leaves of drought stress, pTRV2-0, and pTRV2-*SnRK2.11* plants was significantly greater than that of CK plants, and pTRV2-*SnRK2.11* plants had the highest concentration, which was 41.3% greater than that of CK plants (Figure 3D).

After 7 d of drought stress treatment, the PSII of the leaves of pTRV2-0 and pTRV2-*SnRK2.11* plants was significantly lower than that of CK plants and was 37% and 41.9% less than that of CK plants, respectively; the decrease in drought stress plants was not significant (Figure 3E). In addition, the Fv/Fm of the leaves of pTRV2-*SnRK2.11* plants was significantly less than that of CK plants and was 11.4% less than that of CK plants (Figure 3F). qP was significantly lower in pTRV2-*SnRK2.11* plant leaves than in CK plants, decreasing by 18.7% (Figure 3G), and NPQ was 70.6% greater than that of CK (Figure 3H). These results indicate that the *CsSnRK2.11*-silencing plants reduced photosynthetic capacity under drought stress.

### 2.4. Effects of Drought Stress on Oxidative Damage and Antioxidant Activity of Plants

Oxidative stress is the main mechanism by which plants are damaged under adverse conditions. Antioxidant enzymes scavenge peroxides and reduce oxidative damage. As shown in Figure 4, SOD activity significantly increased in drought stress plants and decreased in pTRV2-0 and pTRV2-*SnRK2.11* plants (Figure 4A). The POD activity of drought stress, pTRV2-0, and pTRV2-*SnRK2.11* plants was significantly higher than that of CK plants. POD activity was highest in pTRV2-*SnRK2.11* plants (Figure 4B). CAT activity of drought stress and pTRV2-0 plants was significantly higher than that of CK plants, and pTRV2-*SnRK2.11* plants had values that were significantly lower than that of CK plants (Figure 4C). The H_2_O_2_ content in drought stress, pTRV2-0, and pTRV2-*SnRK2.11* plants was significantly greater than that of CK plants, and the pTRV2-0 plants showed the greatest increase in H_2_O_2_ content (Figure 4D). The MDA content of plants subjected to drought stress, pTRV2-0, and pTRV2-*SnRK2.11* plants was significantly higher than that of CK plants, with the *SnRK2.11*-silencing plants having the highest content (Figure 4E). The proline content of drought stress, pTRV2-0, and pTRV2-*SnRK2.11* plants was significantly higher than that of CK plants, with pTRV2-*SnRK2.11*-silencing plants having up to more than five-fold higher content than that of CK plants, and drought stress and pTRV2-0 plants having a content up to three-fold higher than that of CK plants (Figure 4F). These results suggest a close relationship between *CsSnRK2.11* silencing and antioxidant damage in plants.

After drought stress treatment for 7 d, the leaves were stained to observe the presence and distribution of malondialdehyde, hydrogen peroxide, and superoxide anions. As shown in Figure 4G, the MDA staining results showed that the colours of drought stress, pTRV2-0, and pTRV2-*SnRK2.11* plants significantly differed from those of CK plants, with pTRV2-*SnRK2.11* plants having the darkest colour, indicating a high accumulation of MDA. The H_2_O_2_ staining distribution showed that the colour of drought stress, pTRV2-0, and pTRV2-*SnRK2.11* plants was significantly different from that of CK plants, with drought stress and pTRV2-*SnRK2.11* plants having a slightly lighter yellow-brown colour than that of CK plants, and pTRV2-0 plants having a slightly darker yellow-brown colour than that of CK plants, indicating that the leaves were severely damaged and accumulated H_2_O_2_ under drought stress. Superoxide anion distribution showed that the colour of drought stress, pTRV2-0, and pTRV2-*SnRK2.11* plants significantly differed from that of CK plants. In the three treatments, blue precipitates were mainly present around the main leaf veins and in small amounts at the leaf margins, indicating that the accumulated superoxide anion content decreased.

### 2.5. Differentially Expressed Gene Analysis

The expression difference ploidy|log2FoldChange| > 1 and significance *p*-value < 0.05 were used as the screening criteria pairs for differential gene expression, and DEGs were screened for the control and silenced groups. CK-vs-SnRK2.11 (control vs. silenced group) was used to represent the differential gene expression set (Figure 5). After 7 d of drought stress, 534 genes were differentially expressed between the control and silenced groups: 311 upregulated and 223 downregulated genes.

### 2.6. Differential Gene GO Functional Enrichment Analysis

As shown in Figure 6. The cellular component (CC) term comprised the MCM complex (mini-chromosome maintenance, which serves as a licensing factor for DNA replication to make sure that genomic DNA is replicated completely and accurately once during the S phase in a single cell cycle), transcription regulator complex, host cellular components, host cell parts, host intracellular parts, host intracellular organelles, host intracellular-membrane-bound organelles, viral factories, nuclear viral factories, and host cell nuclei. Except for DEGs in the transcription regulator complex, all DEGs genes in the CC term were downregulated. This result suggests that *SnRK2.11* gene silencing might cause severe damage to plant cellular components under drought stress.

The molecular function (MF) term comprised heme binding, tetrapyrrole binding, iron ion binding, oxidoreductase activity, (oxidoreductase activity, acting on paired donors, incorporation or reduction of molecular oxygen), cofactor binding, UDP-glucosyltransferase activity, DNA-binding transcription factor activity, transcription regulator activity, and L-threonine ammonialyase activity. The number of upregulated expressed genes of DEGs in the MF term was significantly higher than that of the downregulated genes, indicating that *SnRK2.11* gene silencing might be sensitive to drought stress in plants.

The biological process (BP) term comprises DNA replication initiation, the oxidation–reduction process, DNA-dependent DNA replication, DNA replication, the DNA metabolic process, the cellulose biosynthetic process, the UDP-glucose metabolic process, the cellulose metabolic process, response to wounding, and response to stress. The number of DEGs in the MF term was mostly downregulated and significantly higher than the number of upregulated genes, indicating that *SnRK2.11* gene silencing might affect plant biological processes under drought stress.

### 2.7. Annotation Analysis of Differential Gene KEGG

KEGG enrichment analysis was performed for metabolic pathways in which the DEGs were differentially expressed (Figure 7). Four pathways—plant hormone signal transduction, phenylpropanoid biosynthesis, the MAPK signalling pathway, and starch and sucrose metabolism—were significantly enriched in DEGs, but not to a high degree. Most DEGs in the metabolic pathways of plant hormone signal transduction and starch and sucrose metabolism were upregulated. Phenylpropanoid biosynthesis and the MAPK signalling pathway differential genes were mostly downregulated. DNA replication and carotenoid biosynthesis pathways were highly enriched for DEGs but not significantly. In conclusion, *SnRK2.11*-silencing plants under drought stress may affect metabolic pathways such as phytohormone signalling, MAPK signalling, starch and sucrose metabolism, and carotenoid biosynthesis. Seedling response mechanisms under stress are affected, and organismal defences and adaptations are weakened.

### 2.8. Analysis of DEGs in the MAPK Signalling Pathway, Plant Hormone Signalling Pathways, and Carotenoid Biosynthetic Pathway

The DEGs in the MAPK signalling pathway are shown in Figure 8A. Three genes were upregulated, and four genes were downregulated. Three of the upregulated and one of the downregulated genes were annotated to the ABA signalling pathway of the salt/drought/osmotic stress pathway, namely the protein phosphatase 2C family genes (CsaV3_1G034580), CsPP2C20 (CsaV3_3G016530), CsPP2C23 (CsaV3_3G027970), and catalase (CsaV3_6G031490). This result indicates that the target gene-silenced plants affected the MAPK signalling pathway-ABA signalling pathway-coupled signalling regulatory pathway under drought stress. The other three downregulated genes involved in pathogen infection, pathogen attachment, and ethylene signalling pathways were pathogenesis-related protein 1 (CsaV3_2G012870), ethylene-responsive transcription factor 1 B (CsaV3_3G012170), and ethylene-responsive transcription factor 1B like (CsaV3_2G012870) coding genes. This result suggests that under stress from TRV and plant fusion, effective silencing of the *SnRK2.11* gene affects the MAPK signalling pathway, which may be negatively regulated to affect the plant.

DEGs in the plant hormone signalling pathway are shown in Figure 8B. There were 13 upregulated and 5 downregulated genes. Seven of the upregulated genes were annotated to the tryptophan metabolism pathway, namely genes encoding auxin-responsive proteins (CsaV3_4G026200, CsaV3_7G027610, CsaV3_3G046370, CsaV3_3G026270, and CsaV3_7G022810), the gene encoding indole-3-acetic acid-amido synthetase GH3.6 (CsaV3_3G023380), and auxin-responsive protein SAUR21-like (CsaV3_2G015520). The diterpene biosynthetic pathway was annotated as an upregulated gene encoding an F-box protein GID2-like (CsaV3_2G026400). Four upregulated genes and one downregulated gene of ABA signalling were annotated to the carotenoid biosynthetic pathway, namely the genes encoding protein phosphatase 2C (CsaV3_1G034580, CsaV3_3G027970, and CsaV3_3G016530), ABSCISIC ACID-INSENSITIVE 5-like (CsaV3_4G028620), and the gene encoding ABSCISIC ACID-INSENSITIVE 5-like protein 2 (CsaV3_1G041950). The cysteine and methionine metabolic pathways were annotated as two downregulated genes encoding ethylene-responsive transcription factor 1B-like (CsaV3_1G041950) and ethylene-responsive transcription factor 1 B (CsaV3_2G012870). One downregulated gene encoding a kinase family protein was annotated in the brassinosteroid biosynthesis pathway (CsaV3_4G037140). The phenylalanine metabolic pathway was annotated to one upregulated gene and one downregulated gene encoding transcription factor-like and pathogenesis-related protein 1 (CsaV3_4G004890, CsaV3_7G007620). Therefore, *SnRK2.11* gene silencing in plants under drought stress may affect phytohormone signalling pathways, especially those that regulate plant growth and response to stress.

The DEGs in the carotenoid biosynthetic pathway are shown in Figure 8C. Three genes were upregulated, and two genes were downregulated. The three upregulated genes encoded 9-cis-epoxy carotenoid dioxygenase (CsaV3_2G026690) and ABA 8’-hydroxylase (CsaV3_4G007140 and CsaV3_4G007760). The two downregulated genes were octahydrolycopene synthase (CsaV3_4G023380) and β-carotene hydroxylase (CsaV3_5G002520). This result indicates that silencing the target gene affected the carotenoid synthesis pathway under drought stress.

### 2.9. qRT-PCR Validation of DEGs

The top 10 genes with the most significant differences among the DEGs were used for qRT-PCR validation (Figure 9). The CsaV3_ 3G018770 verification results were inconsistent with the transcriptome data, and the remaining nine genes were consistent with the transcriptome data (90% consistency), indicating that the transcriptome data are accurate and reliable.

## 3. Discussion

Different *SnRK2* genes are involved in plant responses to various stressors as positive regulatory factors in the ABA signalling pathway, and their regulatory effects vary among the members. *SnRK2* overexpression in *Arabidopsis* and *tobacco* has been reported to play an important role in drought resistance. However, there are few reports on using VIGS technology to silence the *SnRK2* gene to explore its drought resistance. In the early stage, using PEG to simulate drought stress, we found that the *SnRK2.11* gene is sensitive to drought stress and has high gene expression. Therefore, we used VIGS technology to insert *SnRK2.11* as a candidate target gene into the pTRV2 vector and transformed cucumber seedlings infected with *Agrobacterium tumefaciens* to silence *SnRK2.11* gene, preliminarily exploring its drought resistance function.

Drought stress can inhibit plant growth parameters, including plant growth, leaf expansion, branch length, root length, root stem ratio, leaf area index, and biomass production [18]. After effectively silencing the target gene, the *SnRK2.11*-silencing plants showed a higher degree of wilting and shorter phenotypes than the other treatments (Figure 1A,B). *SnRK2.11* gene expression was significantly upregulated under drought stress and pTRV2-0 treatment and significantly downregulated under pTRV2-*SnRK2.11* treatment (Figure 1C). Tian et al. [8] found that excessive expression of *TaSnRK2.3* in *Arabidopsis* resulted in longer main roots and more lateral roots under drought stress. This result is similar to our research. When plants are under drought stress, they undergo a series of reactions to resist stress, leading to the reduction of plant respiration and photosynthesis and the weakening of the water-holding capacity of leaves, which is directly reflected by the degradation of chlorophyll, leaf gas exchange parameters, chlorophyll fluorescence parameters, and leaf water content [19,20]. In this study, the photosynthetic pigment content (Figure 2A–D), relative leaf water content (Figure 2E), photosynthetic parameters (Pn, Tr, Gs) (Figure 3A–C), and chlorophyll fluorescence parameters (PSII, Fv/Fm, qP) of *SnRK2.11*-silencing plants were significantly lower than those of other treatments (Figure 3E–G). Ci (Figure 3D) and NPQ (Figure 3H) levels significantly increased. Zhang et al. [21] reported that overexpression of *TaSnRK2.8* in *Arabidopsis* resulted in enhanced tolerance to drought, salt and cold stresses, strengthened cell-membrane stability, significantly reduced osmotic potential, enhanced chlorophyll content, and enhanced PSII activity. This result is consistent with our research findings. Drought leads to a decrease in chlorophyll synthesis, degradation of chlorophyll within the plant, production of ABA, induction of stomatal closure, inhibition of transpiration, water loss, and inhibition of photosynthesis [22]. Therefore, under drought stress, the *SnRK2.11*-silencing plants have reduced water and photosynthetic pigment content, a decreased photosynthetic rate, inhibited transpiration water loss, and reduced photosynthetic and fluorescence parameters. These results indicate that *CsSnRK2.11* plays a crucial role in regulating plant growth, development, and photosynthesis.

Drought stress causes excessive accumulation of ROS, leading to oxidative damage to plant cell membranes. Meanwhile, ROS accelerates cell aging and disintegration, leading to an increase in MDA content in plants [23,24,25]. After drought stress, the relative conductivity (Figure 2F), H_2_O_2_ content (Figure 4D), and MDA content (Figure 4E) of pTRV2-*SnRK2.11* plants significantly increased, indicating that the degree of oxidation and membrane damage in silent plants were more severe. Shao et al. [26] found that after transferring the *MpSnRK2.10* gene into apples, the ROS and MDA contents of the transgenic overexpressed plants under drought treatment were lower than those of the wild-type plants. This result is consistent with our research. Plants can remove ROS through antioxidant enzymes such as SOD, POD, and CAT to maintain normal physiological metabolism. An increase in Pro content reflects the resistance of plants to a certain extent, and its content increases with the prolongation of stress, which thereby maintains the osmotic potential of cells, protecting the stability and integrity of membranes, and effectively scavenges ROS [27,28,29,30]. In our study, pTRV2-*SnRK2.11* silencing reduced SOD (Figure 4A) and CAT (Figure 4C) activities and the significantly increased POD activity (Figure 4B) and content of Pro (Figure 4F) under drought stress. Wang et al. [31] found that after silencing the *SLB3* gene in tomatoes, SOD activity decreased and Pro and MDA contents and POD activity increased. This result is consistent with our research. SOD has been shown to increase sharply under mild drought stress, and SOD activity decreases or becomes inactivated when drought stress is too rapid or severe [32,33]. POD and CAT act synergistically with SOD not only by decomposing H_2_O_2_ into H_2_O and O_2_, but also by participating in chlorophyll degradation and ROS production [31]. Therefore, *SnRK2.11*-silencing plants are severely stressed, leading to reduced SOD and CAT activities and excessive membrane lipid oxidation, which increases the degree of stress in silenced plants. We speculate that *CsSnRK2.11* may be involved in regulating cucumber ROS metabolism.

MDA, DAB, and NBT staining are crucial for detecting the extent of oxidative damage to plant membranes. In this experiment, after 7 d of drought stress, the MDA, H_2_O_2_, and O_2_^−^ contents observed in the leaf staining results of all treatments were higher than those of the CK treatment (Figure 4G). Leaf staining of *SnRK2.11*-silencing plants showed a darker reddish-brown and blue colour of MDA and O_2_^−^ than drought stress. H_2_O_2_ can also be produced by peroxidation of the photorespiratory pathway, polyamine oxidase, copperamine oxidase, and sulphite oxidase under drought stress [34,35]. Therefore, some differences in the H_2_O_2_ staining and assay values may have been caused by differences between plants and different oxidation pathways. These results indicate that MDA and ROS accumulation levels in silenced plants were higher than those under drought stress plants. The expression of the suppressor gene *SnRK2.11* increased cucumber’s sensitivity to drought stress and the degree of stress injury.

Under drought stress, plants first respond to the cellular reception of stress signals which regulate the relevant gene expression and intuitive physiological responses through signalling pathways [36]. RNA-Seq technology has the advantages of high sequencing throughput and low cost; it can be applied to multi-species uncertain gene expression assays, which provides advantages in clarifying information transfer between plants and the environment widely used in stress resistance studies [37]. In this experiment, 534 genes were differentially expressed after 7 days of drought stress: 311 were upregulated and 223 were downregulated. GO functions were classified as MF, BP, and CC (Figure 6). The CC term contained the transcriptional regulatory complex, host cell component, nuclear virus plant, host cell nucleus, and other molecular functions, in which the DEGs of the CC term were mostly downregulated. The MF term contained oxidoreductase activity, transcriptional regulatory activity, L-threonine ammonia lyase activity, and other molecular functions, in which the number of upregulated DEGs in the MF term was significantly higher than that in the number of downregulated genes. The BP term contains DNA replication, response to injury, response to stress, and other molecular functions, in which the number of downregulated DEGs in BP term was significantly higher than that of upregulated DEGs. The annotation analysis of the KEGG pathway enrichment showed that after 7 d of drought stress, the DEGs in cucumber seedling leaves mainly involved four pathways: phytohormone signalling, phenylpropanoid biosynthesis, the MAPK signalling pathway, and starch and sucrose metabolism, which were significantly enriched differentially, but not to a high degree (Figure 7). DNA replication and carotenoid biosynthesis pathways were highly enriched but not significantly enriched. The KEGG pathway enrichment annotation results are consistent with the experimental results of Khan et al. [38]. In this study, phytohormone signalling and MAPK signalling pathway DEGs were mostly associated with ABA signalling. Among these, phytohormone signalling was the pathway with the highest significance. This result is consistent with that reported by Zhang et al. [39]. ABA is an endogenous messenger in plant biotic and abiotic stress responses, forming a coupled ABA–PYR/PYL–PP2C–SnRK2-transcription factor signalling pathway in response to biotic and abiotic stresses [40]. In this study, plants with the silenced gene *SnRK2.11* were DEGs annotated to members of the protein phosphatase 2C (PP2C) family, a key gene in the ABA signalling pathway in phytohormone signalling, and members of the ABF family, a downstream transcription factor. MAPK signalling pathways were similarly annotated as members of the protein phosphatase 2C (PP2C) family. By contrast, the ABA receptor PYR/PYL was not annotated as a differentially expressed gene in either pathway. ABF mediates ABA-induced PP2C gene expression and thus plays a role in the forward and reverse regulation of ABA signalling. ABF-mediated transcriptional upregulation of PP2C and PP2C-mediated inactivation of ABF constitute a tight regulatory loop in ABA signalling [41]. The DEGs’ protein phosphatase 2C and ABF transcription factors annotated in this experiment were the most upregulated. These results suggest that the silencing of *SnRK2.11* gene may affect ABA signalling in response to drought stress. No or a minimal expression of *SnRK2.11* disrupts ABA signalling and inhibits transcription factors that mediate ABA response gene expression.

Phytoene synthase (PSY) and β-carotene hydroxylase are key enzymes in the carotenoid biosynthetic pathway [42]. ABA biosynthesis begins with the cleavage of carotenoid precursors catalysed by 9-cis-epoxy carotenoid dioxygenase (NCED), which is a key regulatory step in the ABA biosynthetic pathway [43]. In this study, DEGs were annotated as genes encoding 9-cis-epoxy carotenoid dioxygenase, octameric lycopene synthase, and β-carotene hydroxylase. Arraes et al. [44] reported that the expression of multiple genes involved in the biosynthesis of plant hormones, such as ethylene and ABA, is upregulated under drought stress, and the response to drought stress can be regulated through the crosstalk network between plant hormone signalling pathways. These results suggest that the silencing of *SnRK2.11* gene may affect the carotenoid biosynthesis pathway in response to drought stress, and ABA biosynthesis may be inhibited.

In summary, these results suggest that silencing the *SnRK2.11* gene affects the response of cucumber seedlings to drought stress and reduces their drought resistance. The *SnRK2.11* gene may be a master regulator involved in the phytohormone signalling, MAPK signalling, and carotenoid synthesis pathways that regulate plant growth and stress responses. Nevertheless, further studies are necessary to determine the mechanism by which *SnRK2.11* responds to drought stress in cucumbers and to elucidate the crosstalk between the pathways mediated by *SnRK2.11*.

## 4. Materials and Methods

### 4.1. Plant Materials

Cucumber seeds (*Cucumis sativus* L. “L306”) purchased from Tianjin Xian You Da Seeds Co., Ltd. (Tianjin, China) were grown in a cavity tray substrate. Seedlings were transplanted after the two true leaves were fully expanded. After transplanting, the seedlings were incubated in an artificial climate chamber at 26 °C for 14 h during the day and 18 °C for 10 h at night, with 250 µmol·m^−2^·s^−1^ of light and 70% relative humidity. Seedlings were watered with Yamazaki cucumber nutrient solution (0.5 mM NH_4_H_2_PO_4_, 2.0 mM Ca(NO_3_)_2_·4 H_2_O, 3.2 mM KNO_3_, 1.0 mM MgSO_4_·7 H_2_O and full-strength trace elements) using the weighing method to maintain the substrate moisture content at 85% of the field holding capacity, and each seedling was irrigated with a consistent amount of nutrient solution: 50 mL every 3 d.

### 4.2. VIGS Vector Construction for CsSnRK2.11 Gene and Preparation of Agrobacterium Suspension

The CDS sequence of *CsSnRK2.11* gene was sent to Shaanxi Boride Biotechnology Co., Ltd. (Xi’an, China). The target was designed and synthesised by the manufacturer and cloned into the VIGS vector (pTRV2). Subsequently, the target was transformed into Agrobacterium GV3101 and amplified for validation (Appendix A lists the identification primers). Next, we correctly identified the pTRV2-*CsSnRK2.11*, pTRV1, and pTRV2-0 bacterial solutions inoculated on LB solid medium containing resistance (Kan and Rif) and cultivated them at 28 °C until the colonies grew. A single colony was selected, and LB liquid medium containing resistance (Kan and Rif) was used to expand the culture. This bacterial solution was cultivated until it was approximately 0.8 nm of OD_600_ nm by using IM induction medium to continue the culture. Bacterial solution: IM = 1:25 until the appropriate concentration of OD_600_ nm was reached, approximately 0.6–1, and then, the bacterial body was collected by centrifugation. The mycelia were resuspended in the same volume of the MES suspension (25 mL); the solution was centrifuged again to collect the mycelia. The mycelia were resuspended in 1/2 the volume of MES. At the end of the suspension process, 25 μL acetosyringone was added to the pTRV1 tube to a final concentration of 400 μmol/L. Next, pTRV1: pTRV2-0 and pTRV1: pTRV2-*CsSnRK2.11* were mixed at a ratio of 1:1 and placed in the dark at room temperature for 3 h for injection.

### 4.3. Plant Infection

Plant infections were induced by syringe injection. During injection, a needle was used to create a small hole in the cotyledons, and a needle-free syringe was used to inject the bacterial solution into the wound until the cotyledons were completely waterlogged. After infection, the plants were cultured under dark conditions at 22 °C for 48 h. When the seedlings grew to two leaves and one heart, samples were collected, frozen in liquid nitrogen, and stored at −80 °C for future use. Next, the silencing efficiency was detected by qRT-PCR.

### 4.4. Silencing Efficiency Assay and Seedling Drought Treatment

A plant RNA extraction kit (Beijing, Tiangen, China) was used to extract total RNA. Synthesis of cDNA was achieved using the fastking cDNA dispersion RT supermax kit (Beijing, Tiangen, China), using 2 µL RNA as a template. The primers were designed using gene CDS sequences. The SYBR Green kit was used for quantitative fluorescence analysis. The total volume of the reaction system was 20 µL:2 µL of cDNA template, 10 µL of 2 × SuperReal PreMix Plus, 0.6 µL of forward and reverse primers, 0.4 µL of 50 × ROX Reference Dye, and 6.4 µL of ddH_2_O. A LightCycle^®^ 480 II real-time fluorescence quantitative PCR instrument was used for the qRT-PCR analysis. Each qRT-PCR was performed in triplicate. The cucumber internal reference gene (DQ115883) was used as a control (Appendix A lists the primer sequences). Reaction conditions: pre-denaturation at 95 °C for 15 min, denaturation at 95 °C for 10 s for 40 cycles, and annealing at 60 °C for 25 s. Relative gene expression was calculated using 2^−ΔΔCt^ [45].

After the silencing efficiency assay, unimpregnated plants (labelled drought stress), pTRV2-0 null plants, and target gene pTRV2-*SnRK2.11*-silencing plants were collected, and watering was stopped for 7 d. Plants were photographed and observed, and those with significant differences were selected to determine the relevant indexes. The substrate water content was weighed; it decreased to 40% of the field holding capacity 7 d after watering was stopped.

### 4.5. Determination of Chlorophyll Content

Chlorophyll content was determined by spectrophotometry after extraction with 95% ethanol. Absorbance was measured at 440, 663, and 645 nm, according to Lichtenthaler and Wellburn’s method [46].

### 4.6. Determination of Leaf Moisture Content and Relative Conductivity

The relative water concentration (RWC) was measured according to a previously described method [47]. Relative electrical conductivity (REC) was determined according to the method of Gao Junfeng [48].

### 4.7. Measurement of Photosynthetic Parameters and Chlorophyll Fluorescence Parameters

Three plants from each treatment were sampled, and the photosynthetic parameters of the three leaves were measured. A convenient photosynthetic fluorometer (Yaxin-1105; Beijing Yaxin RI Technology Co., Ltd., Beijing, China) was used. The time of the measurement was 09:00. The parameters were 75% relative humidity and a light quantum flux of 1200 µmol·m^−2^·s^−1^. The net photosynthetic rate (Pn), intercellular carbon dioxide concentration (Ci), stomatal conductance (Gs), and transpiration rate (Tr) were measured at room temperature [49].

The leaves of each plant were incubated in the dark for 30 min. Maximum photochemical efficiency (PSII), actual photochemical efficiency (PSII), non-photochemical burst coefficient (NPQ), and photochemical burst coefficient (qP) were determined using an IMAGIN-PAM chlorophyll fluorescence imager (Germany) [50,51]. The parameters were as follows: detection light, 0.1 µmol·m^−2^·s^−1^; photochemical intensity, 81 µmol·m^−2^·s^−1^; saturated pulsed light, 2700 µmol·m^−2^·s^−1^; pulse time, 0.8 s; and interval, 20 s.

### 4.8. Determination of Physiological Indexes (SOD, POD, CAT, H_2_O_2,_ Proline, and MDA)

For SOD, POD, CAT, and H_2_O_2_, we used assay kits (Solarbio, Beijing, China) according to the manufacturer’s instructions. To measure lipid peroxidation, we employed the thiobarbituric acid (TBA) protocol to determine the MDA produced, using methods described in the literature with minor modifications [52,53]. Pro content was determined using the sulfosalicylic acid method: we weighed 0.1 g of the sample, added 1 mL of the extract, and ground the mixture to homogenate it in an ice bath. The extract was shaken in boiling water for 10 min; the supernatant was collected and added to the reagent; and absorbance was recorded at 520 nm for assay tube A, standard tube A, and blank tube A.

### 4.9. Malondialdehyde, Hydrogen Peroxide, Superoxide Anion Staining Observation

The leaves of CK, drought stress, pTRV2-0, and pTRV2-*SnRK2.11* plants were collected after 7 days of drought stress treatment. Malondialdehyde was performed using thiobarbituric-acid-reactive substances [TBARS] tissue-staining method according to the previously described method [54]. Superoxide anion was detected by nitroblue tetrazolium (NBT) staining according to previously described methods [55]. In addition, leaf samples were stained with 3,3′-diaminobenzidine (DAB) to detect hydrogen peroxide [56,57].

### 4.10. RNA Extraction, Detection, Library Construction, and Transcriptome Sequencing

The cells were cultured, treated, and sampled as aforementioned. After sampling, transcriptome sequencing (tagged CK-vs-SnRK2.11) was performed on the drought-treated control and silent group samples by Chengdu Panomic Biotechnology Co (Chengdu, China). RNA was extracted using the RNA Prep Pure Polysaccharide Plant Total RNA Extraction kit (Tiangen, Beijing, China). RNA was purified using TRIzol reagent. mRNA with a polyA structure in the total RNA was enriched using oligo (dT) magnetic beads, and ion interruption was used to interrupt the RNA into fragments of approximately 300 bp in length. RNA integrity was determined via RNA-specific agarose gel electrophoresis by using an Agilent 2100 Bioanalyzer. The first strand of cDNA was synthesised with 6-base random primers and reverse transcriptase using RNA as a template. The second strand of cDNA was synthesised using the first strand cDNA as a template. After library construction, library fragment enrichment was performed using PCR amplification, followed by library selection based on fragment size (450 bp). Next, the libraries were quality checked using an Agilent 2100 Bioanalyzer, and the total and effective library concentrations were tested. Libraries containing different index sequences (each sample plus different indexes and, finally, the next data point of each sample according to the index) were mixed proportionally according to the effective concentration of the library and the amount of data required for the library. Paired-end (PE) sequencing of these libraries was performed using Next-Generation Sequencing (NGS) technology based on the Illumina sequencing platform.

The sequenced downstream data containing impurities (Raw Data) were filtered, and low-quality data were removed to obtain high-quality sequences (Clean Data). The reference gene set was ChineseLong_genome_v3, and the results were used to calculate the expression of each gene. The samples were analysed for expression differences, enrichment, and clustering based on the gene expression results. Transcript sequences were reduced by splicing the reads into pairs.

### 4.11. qRT-PCR Validation of DEGs

To ensure the accuracy and reliability of the transcriptome data, we selected the 10 most significant DEGs for qRT-PCR validation. The samples were obtained from the company, and the RNA reverse transcription and fluorescence quantification methods used were the same as those aforementioned. Primer 5.0 was used to design gene primers synthesised by Beijing Prime Tech Biotechnology Co. (Appendix A lists the primer sequences).

### 4.12. Statistical Analysis

Data were subjected to one-way ANOVA with Duncan’s multiple range test (*p* < 0.05) using the statistical software SPSS 20.0. Histograms of the expression of interest were constructed using Origin 9.0, and the values of the final plotted graphs represent the means of three replicates.

## 5. Conclusions

*CsSnRK2.11*-silencing plants were obtained using the VIGS technique, and the plants dwarfed and wilted faster after drought treatment than before. Photosynthetic parameters (Pn, Tr, and Gs), chlorophyll fluorescence parameters (PSII, Fv/Fm, and qP), relative leaf water content, chlorophyll content, and the activities of the antioxidant enzymes SOD and CAT were significantly decreased. Ci, NPQ, MDA, and Pro levels were significantly increased. RNA-seq analysis revealed 534 DEGs: 311 upregulated and 223 downregulated. GO functional annotation analysis showed that cell components, molecular functions, and molecular functions of biological processes are important mechanisms in plant responses to stress. KEGG pathway enrichment annotation analysis showed that DEGs were significantly enriched in plant hormone signal transduction; it also showed that the MAPK signalling pathway and the carotenoid synthesis pathway were related to ABA. In summary, this study analysed the function of *CsSnRK2.11* under drought stress at the physiological, biochemical, and transcriptome levels, providing a theoretical basis for studying the mechanism of the *SnRK2* gene in cucumber drought resistance.

## Figures and Tables

**Figure 1 ijms-24-15761-f001:**
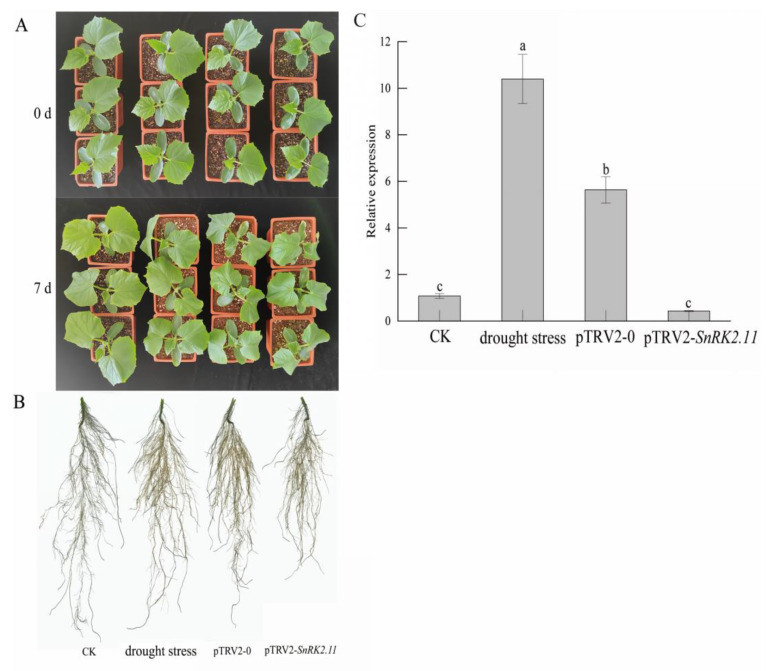
Phenotypic observation of cucumber seedlings and relative expression level of *CsSnRK2.11* under drought stress. CK: not-drought control plants; drought stress: unimpregnated control plants; pTRV2-0: empty vector plants; pTRV2-*SnRK2.11*: *SnRK2.11*-silencing plants. Plant phenotype (**A**) and root size (**B**) of CK, drought stress, pTRV2-0, and pTRV2-*SnRK2.11* plants after 0 and 7 d of drought stress. The relative expression level of *CsSnRK2.11* under drought stress (**C**). The data are presented as the means ± SE (*n* = 3). Different letters indicate significant differences at *p* < 0.05.

**Figure 2 ijms-24-15761-f002:**
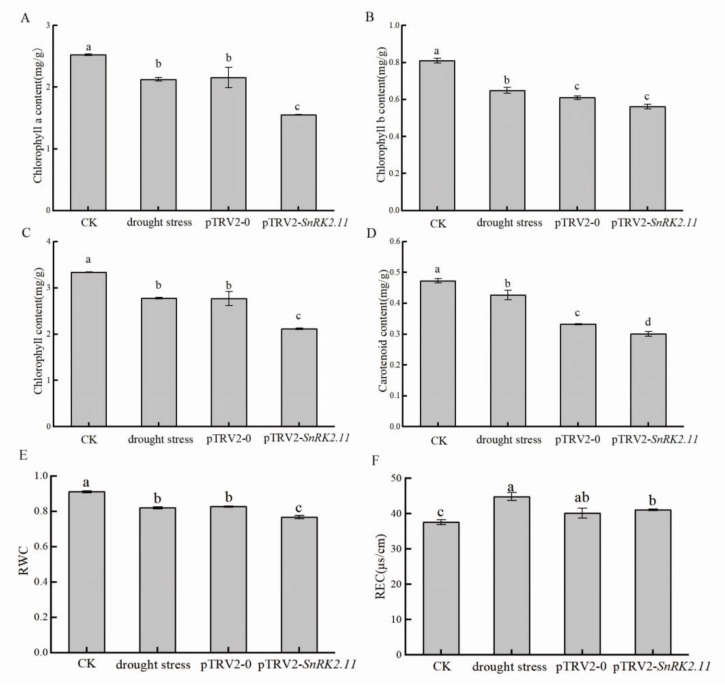
Effect of drought stress on chlorophyll content, relative water content and relative electrical conductivity of cucumber. CK: not-drought control plants; drought stress: unimpregnated control plants; pTRV2-0: empty vector plants; pTRV2-*SnRK2.11*: *SnRK2.11*-silencing plants. (**A**) Chlorophyll a content; (**B**) Chlorophyll b content; (**C**) Chlorophyll content; (**D**) Carotenoid content; (**E**) RWC (Relative leaf water content); (**F**) REC (relative conductivity of leaves). The data are presented as the means ± SE (*n* = 3). Different letters indicate significant differences at *p* < 0.05.

**Figure 3 ijms-24-15761-f003:**
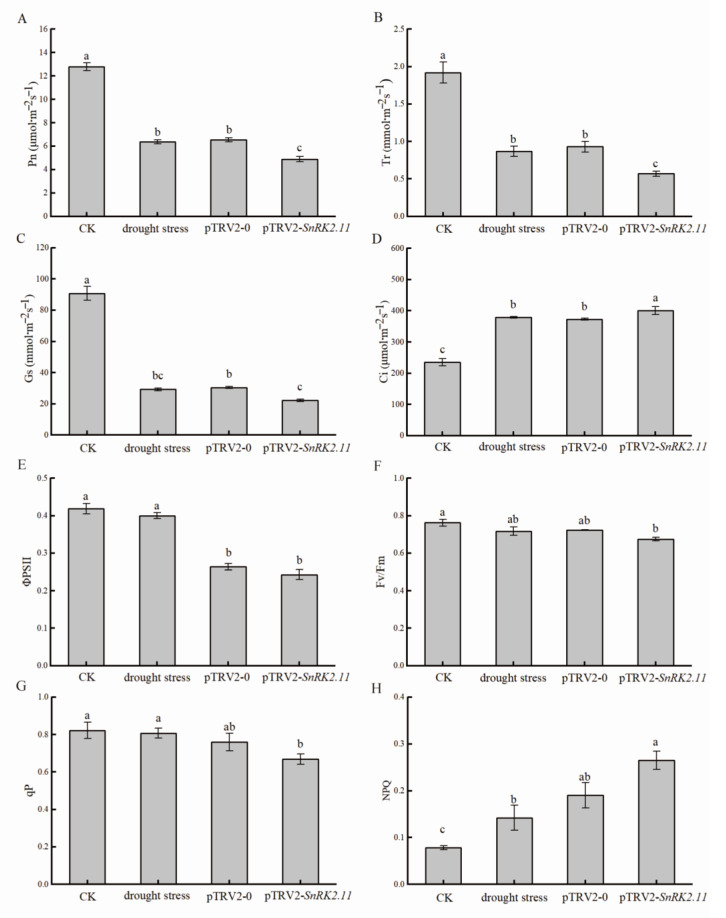
Effects of drought stress on gas exchange parameters and chlorophyll fluorescence parameters of plant leaves. CK: not-drought control plants; drought stress: unimpregnated control plants; pTRV2-0: empty vector plants; pTRV2-*SnRK2.11*: *SnRK2.11*-silencing plants. (**A**) Pn (Net photosynthetic rate); (**B**) Tr (Transpiration rate); (**C**) Gs (Stomatal conductance); (**D**) Ci (Intercellular carbon dioxide concentration); (**E**) ΦPSII (Actual photochemical efficiency); (**F**) Fv/Fm (Maximum photochemical efficiency); (**G**) qP (Photochemical quenching coefficient); (**H**) NPQ (Non-photochemical quenching coefficient). The data are presented as the means ± SE (*n* = 3). Different letters indicate significant differences at *p* < 0.05.

**Figure 4 ijms-24-15761-f004:**
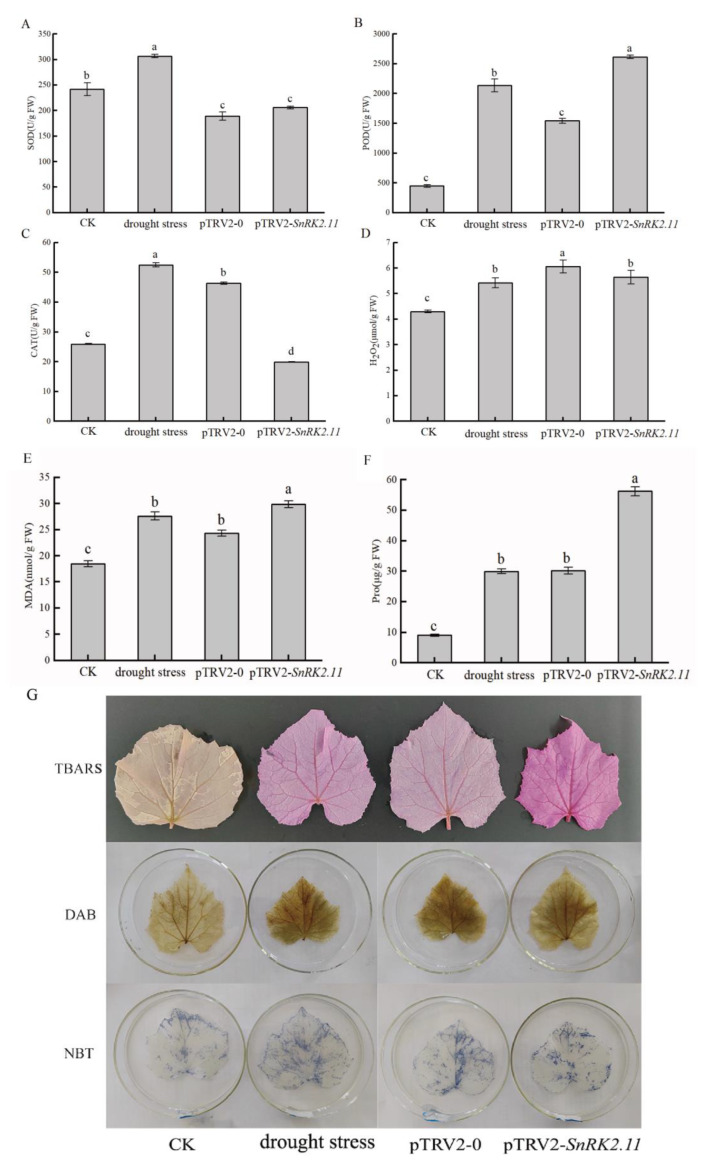
Effects of drought stress on (**A**) SOD, (**B**) POD, (**C**) CAT activities, (**D**) H_2_O_2_ and (**E**) MDA and (**F**) Pro contents in cucumber leaves. (**G**) Observation of leaf staining under drought stress. CK: not-drought control plants; drought stress: unimpregnated control plants; pTRV2-0: empty vector plants; pTRV2-*SnRK2.11*: *SnRK2.11*-silencing plants. TBARS (malondialdehyde staining observation); DAB (hydrogen peroxide staining observation); NBT (superoxide anion staining observation). The data are presented as the means ± SE (*n* = 3). Different letters indicate significant differences at *p* < 0.05.

**Figure 5 ijms-24-15761-f005:**
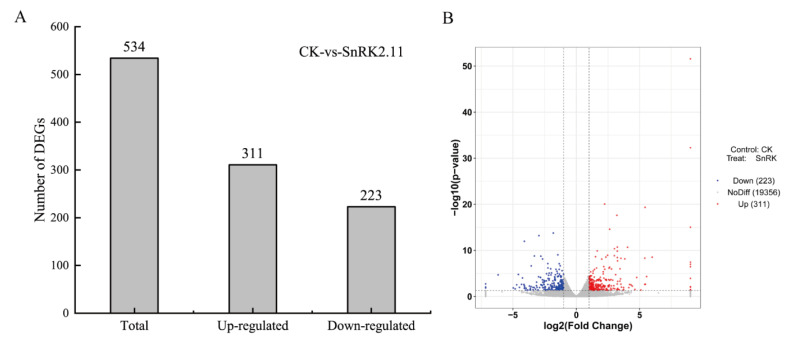
Number of differential gene expression under drought stress. (**A**) Statistics of the number of differential genes; (**B**) Volcano map of differentially expressed genes. The *x* and *y* axis denote the log 2-fold change and −log10 (*p*-value). Red, gray, and blue correspond to up-regulated, unaltered, and down-regulated gene expression, respectively.

**Figure 6 ijms-24-15761-f006:**
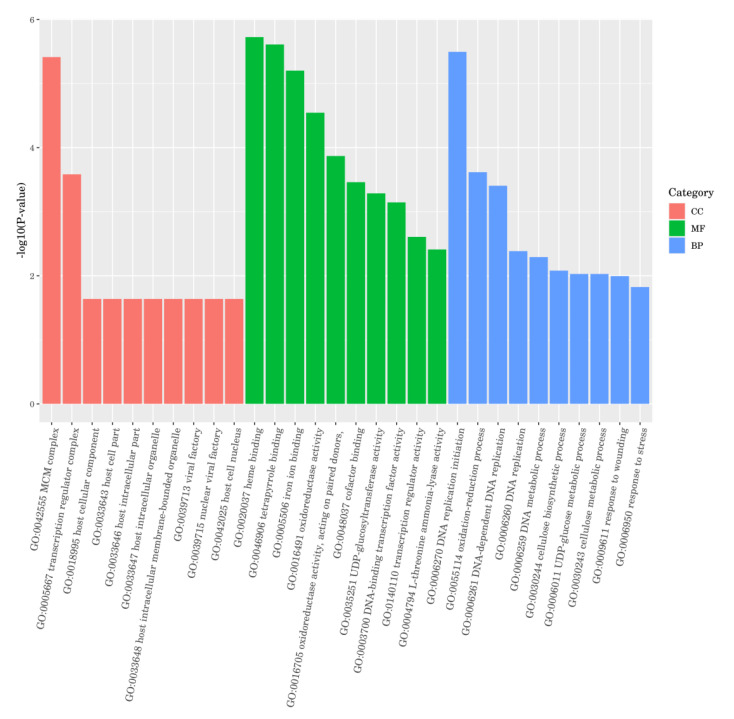
Analysis of GO function enrichment of differential genes. The *x*-axis indicates the GO classification, and the *y*-axis indicates the number of genes in each classification. The red represents cellular component, green represents molecular function, blue represents biological process.

**Figure 7 ijms-24-15761-f007:**
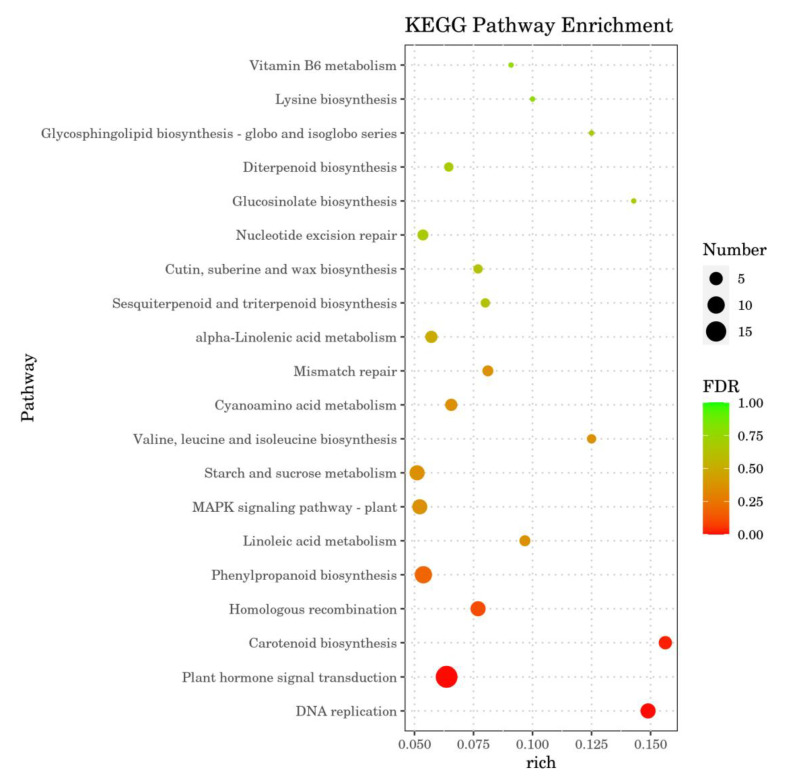
Differential gene KEGG analysis. The *x*-axis shows the rich factor, and the *y*-axis shows the top 20 KEGG pathways. FDR: also known as False Discovery Rate, is obtained by correcting for the *p*-value of significant differences in transcriptome data, and ultimately FDR was used as the key indicator for differential expression gene screening. The bigger size of spot, the more DEGs were enriched.

**Figure 8 ijms-24-15761-f008:**
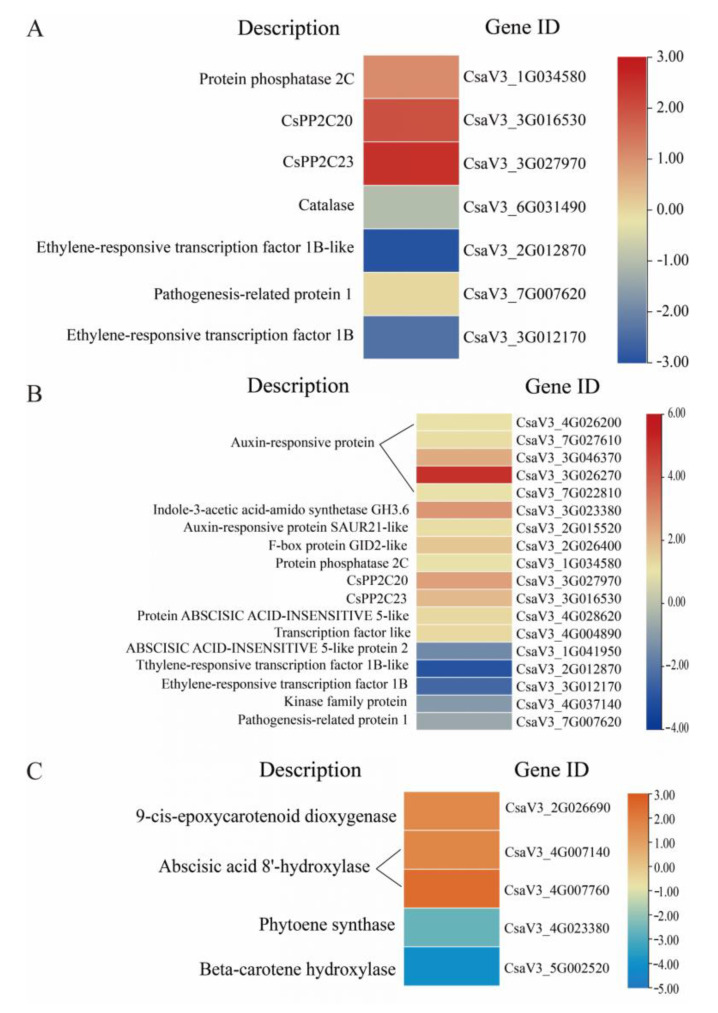
Analysis of differentially expressed genes in the (**A**) MAPK signaling pathway, (**B**) plant hormone signaling pathways, and (**C**) carotenoid biosynthetic pathway.

**Figure 9 ijms-24-15761-f009:**
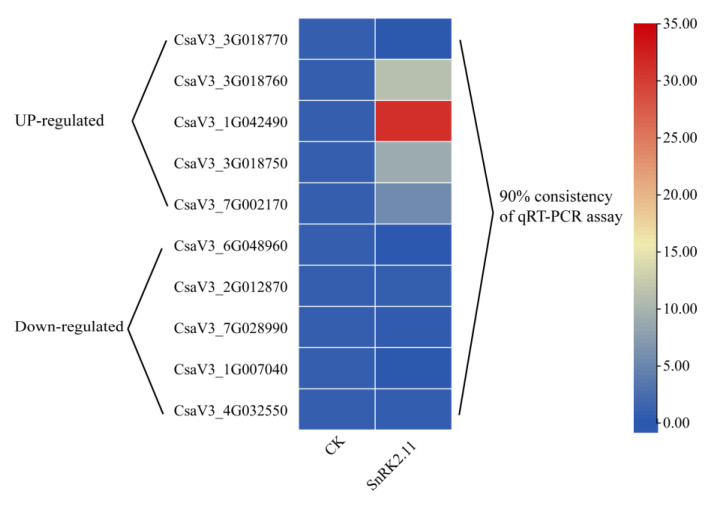
Verification of differentially expressed genes by qRT-PCR.

## Data Availability

All data generated during this study are included in this published article and its Appendix A.

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
