# Peer review of "Silencing the CsSnRK2.11 Gene Decreases Drought Tolerance of Cucumis sativus L."

_ijms, 2023, doi:10.3390/ijms242115761_

Round 1
Reviewer 1 Report
Comments and Suggestions for Authors
The paper focusses on significance of SnRK2.11 in cucumber plants under drought stress condition. The gene is member of SnRK2 family which are serine/threonine kinases generally involved in plant response to abiotic stresses. Using ViGS technology the authors silenced SnRK2.11gene in cucumber cotyledons and then analyzed effects of drought in CK, drought stress, pTRV2-0, and pTRV2-SnRK2.11 plants on antioxidant enzyme activity (SOD, POD, CAT) activities, gas exchange and chlorophyll fluorescence parameters of plant leaves, chlorophyll content, relative water content and relative electrical conductivity of cucumber. It was clearly proved that CsSnRK2.11 gene play key role in drought tolerance of cucumber. The research was methodically well planned and the presented results are beyond doubt, they were well discussed and interpreted. The study provides new knowledge about the importance of CsSnRK2.11. I have no objections to the work, but applied abbreviations (CK plants, VIGS) should be clarified
Author Response
The paper focusses on significance of SnRK2.11 in cucumber plants under drought stress condition. The gene is member of SnRK2 family which are serine/threonine kinases generally involved in plant response to abiotic stresses. Using VIGS technology the authors silenced SnRK2.11 gene in cucumber cotyledons and then analyzed effects of drought in CK, drought stress, pTRV2-0, and pTRV2-SnRK2.11 plants on antioxidant enzyme activity (SOD, POD, CAT) activities, gas exchange and chlorophyll fluorescence parameters of plant leaves, chlorophyll content, relative water content and relative electrical conductivity of cucumber. It was clearly proved that CsSnRK2.11 gene play key role in drought tolerance of cucumber. The research was methodically well planned and the presented results are beyond doubt, they were well discussed and interpreted. The study provides new knowledge about the importance of CsSnRK2.11. I have no objections to the work, but applied abbreviations (CK plants, VIGS) should be clarified.
Response: Thank you for pointing this out. We agree with this comment. The original manuscript has been explained VIGS at L81-L92 and explained CK plants in all legends.
Reviewer 2 Report
Comments and Suggestions for Authors
The submitted manuscript studying the function of CsSnRK2.11 gene in drought stress using Virus- Induced Gene-Silencing (VIGS) technology and analyzes the transcriptome data, offering valuable insights into the mechanism of the SnRK2 gene in increasing drought tolerance in cucumbers.
It should be emphasized the large volume of work and experimental data obtained and analyzed, the interdisciplinarity of the work involving studies of physiology and cell biochemistry as well as molecular genetics. Although in general appearance the manuscript offers a lot of valuable information, there are some mistakes and requirements that must be considered by the authors. In this regard, the following comments are requested to be addressed by the authors:
TITLE - it is preferable to use in the title, and most often in the text, the scientific name of the studied species.
ABSTRACT
ü there are problems with the use of abbreviations for which no explanation is given,
ü the hypothesis that generated the research is not clearly presented.
ü Line 12- It's not about "vegetable development" but about vegetable growth.
KEYWORDS
- the same linguistic terms are used as in the text, or logically keywors aims to use complementary terms to the title, to help a better understanding of the studied topic.
INTRODUCTION
§ The "state of the art" or the "level of knowledge" on the topic to be researched is presented quite narrowly, without an in-depth approach and analysis.
§ The starting hypothesis and the objectives pursued are not clearly explained...
§ line 55 - "studies have shown" – which studies ??? there are no references for these...
§ lines 78-80 - "In this experiment, we studied the function of CsSnRK2.11 in drought stress using VIGS technology and analysed the transcriptome data"...WHY?
Results
They are detailed and quite clearly presented, and the tables and images part contains enough descriptive details, although the size and resolution of some images could be slightly improved.
Discussion
The first two paragraphs (lines 303-320) have nothing to look for here, they are results and methods. In fact, there are a few more phrases that should be moved to the results...
Line 311- Agrobacterium tumefaciens in italic.
I think that the entire discussion chapter should be reanalyzed, better structured and with the argumentation component of the results through comparisons with other complementary research, which would also lead to the improvement of the list of references...
Materials and Methods
ü Cucumber variety "L306"- a few words and the manufacturer,
ü "Yamazaki cucumber nutrient solution" – a brief characteristics description and manufacturer,
ü For the analysis and evaluation methods descriptions, I recommend to describe in detail the new (original) methods and/or those to which you contributed with changes. In case of established/known methods, post the bibliographic reference (authors / standards used), it will save time and space...
The manuscript needs a detailed analysis and recommended improvements.
Good luck!
Author Response
The submitted manuscript studying the function of CsSnRK2.11 gene in drought stress using Virus-Induced Gene-Silencing (VIGS) technology and analyzes the transcriptome data, offering valuable insights into the mechanism of the SnRK2 gene in increasing drought tolerance in cucumbers.
It should be emphasized the large volume of work and experimental data obtained and analyzed, the interdisciplinarity of the work involving studies of physiology and cell biochemistry as well as molecular genetics. Although in general appearance the manuscript offers a lot of valuable information, there are some mistakes and requirements that must be considered by the authors. In this regard, the following comments are requested to be addressed by the authors:
TITLE - it is preferable to use in the title, and most often in the text, the scientific name of the studied species.
Response: Thank you for your reminder. We have revised the title in the original manuscript to “Silencing the CsSnRK2.11 gene decreases drought tolerance of Cucumis sativus L.”
ABSTRACT
- there are problems with the use of abbreviations for which no explanation is given.
Response: Thank you for your reminder. The original manuscript has been provided comments and explained at L16 and L81-L92.
- the hypothesis that generated the research is not clearly presented.
Response: Thank you for your reminder. The original manuscript has been modified at L11-L15. Modify as follows: Drought stress restricts vegetable growth, and abscisic acid plays an important role in its regulation. Sucrose non-fermenting1-related protein kinase 2 (SnRK2) is a key enzyme in regulating ABA signal transduction in plants, and it plays a significant role in response to multiple abiotic stresses. Our previous experiments demonstrated that the SnRK2.11 gene exhibits a significant response to drought stress in cucumbers.
- Line 12- It's not about "vegetable development" but about vegetable growth.
Response: Thank you for your reminder. We have modified it to “vegetable growth” on L11 of the original manuscript.
KEYWORDS
the same linguistic terms are used as in the text, or logically keywords aims to use complementary terms to the title, to help a better understanding of the studied topic.
Response: Thank you for your reminder. We have revised the keywords section in the original manuscript to “VIGS; Drought stress; Photosynthesis; Antioxidant; Transcriptome”.
INTRODUCTION
- The "state of the art" or the "level of knowledge" on the topic to be researched is presented quite narrowly, without an in-depth approach and analysis.
Response: Thank you for your reminder. We conducted an in-depth approach and analysis on the "state of the art" of the research topic on L81 to L92 of the original manuscript.
- The starting hypothesis and the objectives pursued are not clearly explained...
Response: Thank you for your reminder. The original manuscript has been modified at L95-L100.
- line 55 - "studies have shown" – which studies ??? there are no references for these...
Response: Thank you for your reminder. The "studies have shown" in the original manuscript refers to transgenic Arabidopsis overexpressing TaSnRK2.3 showed no significant differences in water retention ability (WRA), free proline content, and chlorophyll from the wild type under normal conditions. Under drought stress, transgenic plants showed higher WRA, proline, and chlorophyll contents than wild-type plants, and the main roots of the former were longer and lateral roots were more abundant than those of the wild type. The “which studies” Locate reference to [8], Tian, S.; Mao, X.; Zhang, H.; Chen, S.; Zhai, C.; Yang, S.; Jing, R. Cloning and characterization of TaSnRK2. 3, a novel SnRK2 gene in common wheat. Journal of experimental botany 2013, 64, 2063-2080.
- lines 78-80 - "In this experiment, we studied the function of CsSnRK2.11 in drought stress using VIGS technology and analysed the transcriptome data"...WHY?
Response: Thank you for your reminder. Because we plan to explore the function of CsSnRK2.11 under drought stress at the physiological, biochemical, and transcriptional levels, providing valuable insights for further understanding the mechanism by which SnRK2 gene enhances cucumber drought resistance. In addition, We have modified it to “In this experiment, we utilized VIGS technology to study the effects of silencing CsSnRK2.11 gene on cucumber plants under drought stress at physiological and biochemical levels, and combined transcriptome data to analyze its function under drought stress, providing insights into the mechanism by which SnRK2 gene enhances cucumber drought resistance.” on L95-100 of the original manuscript.
Results
They are detailed and quite clearly presented, and the tables and images part contains enough descriptive details, although the size and resolution of some images could be slightly improved.
Response: Thank you for your reminder. We have made slight modifications to the image size and resolution in the original manuscript.
Discussion
- The first two paragraphs (lines 303-320) have nothing to look for here, they are results and methods. In fact, there are a few more phrases that should be moved to the results...
Response: Thank you for your reminder. The original manuscript has been modified at L334-371.
- Line 311- Agrobacterium tumefaciens in italic.
Response: Thank you for your reminder. The original manuscript has been modified at L342.
- I think that the entire discussion chapter should be reanalyzed, better structured and with the argumentation component of the results through comparisons with other complementary research, which would also lead to the improvement of the list of references...
Response: Thank you for your reminder. We have reanalyzed the entire discussion chapter to better structure it by comparing it with other supplementary studies.
Materials and Methods
- Cucumber variety "L306"- a few words and the manufacturer.
Response: Thank you for your reminder. We have modified it to “Cucumber seeds (Cucumis sativus L. “L306”) purchased from Tianjin Xian You Da Seeds Co., Ltd (Tianjin, China)” on L474-475 of the original manuscript.
- "Yamazaki cucumber nutrient solution" – a brief characteristics description and manufacturer.
Response: Thank you for your reminder. We have modified it to “Yamazaki cucumber nutrient solution (0.5 mM NH4H2PO4, 2.0 mM Ca(NO3)2·4 H2O, 3.2 mM KNO3, 1.0 mM MgSO4·7 H2O and full-strength trace elements)” on L479-481 of the original manuscript.
- For the analysis and evaluation methods descriptions, I recommend to describe in detail the new (original) methods and/or those to which you contributed with changes. In case of established/known methods, post the bibliographic reference (authors /standards used), it will save time and space...
Response: Thank you for your reminder. The original manuscript has been modified at L535-537 and L564-570.
Reviewer 3 Report
Comments and Suggestions for Authors
The paper “Silencing the CsSnRK2.11gene decreases drought tolerance of cucumber” by Peng Wang et al. showed that reduction of SnRK2.11 gene in cucumber increased sensitivity to drought stresses using VIGS technology. The authors also conducted RNA-seq analysis to identity the DEGs between SnRK2.11-silencing plants and control plants followed by GO functional annotation analysis and KEGG pathway enrichment analysis. The authors found that expression of genes for phytohormone signaling, MAPK signaling and carotenoid biosynthesis pathways, which are related to ABA synthesis and signaling, were enriched. The results will be useful for the researcher in this field. However, there are several concerns. The authors should revise the text.
1. First of all, it was hard to understand the paper because or the lack of explanation. What is VIGS? The authors should mention about it in the introduction or results.
2. results: authors should explain what they did at first. For example, in the first sentence of 2.1 section, the authors abruptly start the explanation of Fig. 1. However, we cannot understand what it pTRV2.0, pTRV2-SnRnRK2.11, and CK. The authors should explain what they are.
This kind of things are applied to all of results. Please explain what you are doing.
3. Figure 1 and its related text.: as mentioned above, I cannot understand pTRV2.0, pTRV2-SnRnRK2.11, and CK. The worth word is drought stress. I think all plants are suffering from drought stress if I’m not misunderstanding. The authors should add the explanation about the experimental condition in the text and legend of the figure 1.
4.Figure 1 C: the authors show the expression of SnRK2 under drought stress. They should analyze the expression under not drought stress.
5. L130: What are Pn, Tr, and Gs? The authors should write the full name of abbreviation when they use them first time.
6. L204. I cannot understand what they are doing. What is MCM?
7. Figure 7. What is FDR?
8. L260: what is TRV?
9. L307. The authors state that “In the early stage, using PEG to stimulated rought stress, we found that the SnRK22.11 gene is,,,”. Is this unpublished experiment? If the authors have data, it is better to show it. if the authors won’t show it, it should be mentioned as (unpublished data), or something.
Author Response
The paper “Silencing the CsSnRK2.11 gene decreases drought tolerance of cucumber” by Peng Wang et al. showed that reduction of SnRK2.11 gene in cucumber increased sensitivity to drought stresses using VIGS technology. The authors also conducted RNA-seq analysis to identity the DEGs between SnRK2.11-silencing plants and control plants followed by GO functional annotation analysis and KEGG pathway enrichment analysis. The authors found that expression of genes for phytohormone signaling, MAPK signaling and carotenoid biosynthesis pathways, which are related to ABA synthesis and signaling, were enriched. The results will be useful for the researcher in this field. However, there are several concerns. The authors should revise the text.
- First of all, it was hard to understand the paper because or the lack of explanation. What is VIGS? The authors should mention about it in the introduction or results.
Response: Thank you for pointing this out. We agree with this comment. The original manuscript has been explained VIGS at L81-L92.
- results: authors should explain what they did at first. For example, in the first sentence of 2.1 section, the authors abruptly start the explanation of Fig. 1. However, we cannot understand what it pTRV2.0, pTRV2-SnRK2.11, and CK. The authors should explain what they are.
Response: Thank you for your reminder. We have explained pTRV2.0, pTRV2-SnRK2.11, and CK in all the legends in the original manuscript.
- Figure 1 and its related text.: as mentioned above, I cannot understand pTRV2.0, pTRV2-SnRK2.11, and CK. The worth word is drought stress. I think all plants are suffering from drought stress if I’m not misunderstanding. The authors should add the explanation about the experimental condition in the text and legend of the figure 1.
Response: Thank you for your reminder. We have explained pTRV2.0, pTRV2-SnRK2.11, and CK and added an explanation of the experimental conditions to all the legends in the original manuscript.
- Figure 1 C: the authors show the expression of SnRK2 under drought stress. They should analyze the expression under not drought stress.
Response: Thank you for your reminder. In Figure 1 C, CK represents the expression under not drought stress.
- L130: What are Pn, Tr, and Gs? The authors should write the full name of abbreviation when they use them first time.
Response: Thank you for your reminder. Pn (Net photosynthetic rate); Tr (Transpiration rate); Gs (Stomatal conductance). Pn, Tr, and Gs have already been explained on L174-175 of the original manuscript legend.
- L204. I cannot understand what they are doing. What is MCM?
Response: Thank you for your reminder. The MCM complex controls the once per cell cycle DNA replication in eukaryotic cells. In a process known as DNA replication licensing, it primes chromatin for DNA replication by binding origins of DNA replication during the late M to early G1 phase of the cell cycle.
- Figure 7. What is FDR?
Response: Thank you for your reminder. FDR, also known as False Discovery Rate, is obtained by correcting for the p-value of significant differences. Due to the fact that differential expression analysis in transcriptome sequencing involves independent statistical hypothesis testing of a large number of gene expression values, there may be false positives. Therefore, in the process of differential expression analysis, the recognized Benjamin Hochberg correction method was used to correct the significance p-value obtained from the original hypothesis testing, and ultimately FDR was used as the key indicator for differential expression gene screening. Generally, FDR<0.01 or 0.05 is taken as the default standard.
- L260: what is TRV?
Response: Thank you for your reminder. TRV (tobacco rattle virus) is a bipartite, positive-strand RNA virus with the TRV1 and TRV2 genomes. It is a widely used, efficient, and persistent viral vector that can mediate gene silencing without causing virus induced symptoms. TRV induced gene silencing (TRV-VIGS) is currently the most widely used type of gene silencing system.
- L307. The authors state that “In the early stage, using PEG to stimulated rought stress, we found that the SnRK22.11 gene is,,,”. Is this unpublished experiment? If the authors have data, it is better to show it. if the authors won’t show it, it should be mentioned as (unpublished data), or something.
Response: Thank you for your reminder. This is a published experiment. In the study by Zilong Wan et al., it was found that the relative expression level of CsSnRK2.11 was most significant under PEG induction. Therefore, we further investigated the function of CsSnRK2.11 under drought stress. Wan, Z.; Luo, S.; Zhang, Z.; Liu, Z.; Qiao, Y.; Gao, X.; Yu, J.; Zhang, G. Identification and expression profile analysis of the SnRK2 gene family in cucumber. PeerJ 2022, 10, e13994.
Round 2
Reviewer 3 Report
Comments and Suggestions for Authors
The authors have addressed most of my concerns.
The authors gave answers to my question about words. However, I think the author should add short explanation about these words, such as MCM, FDR, and TRV in the text.
Author Response
Thank you for your reminder. The original manuscript has been modified.
Modify as follows:
TRV: Explained on L87-89 of the original manuscript
MCM: Explained on L237-239 of the original manuscript
FDR: Explained on L279-281 of the original manuscript